# The Tufts fNIRS Mental Workload Dataset & Benchmark for Brain-Computer Interfaces that Generalize

Zhe Huang[1,*], Liang Wang[1,*], Giles Blaney[2], Christopher Slaughter[3], Devon McKeon[1], Ziyu Zhou[1], Robert J. K. Jacob[1,*], and Michael C. Hughes[1,*]

[1]Dept. of Computer Science, Tufts University
[2]Dept. of Biomedical Engineering, Tufts University
[3]Dept. of Computer Engineering, Univ. of Maryland Baltimore County
[*]Lead authors ZH & LW contributed equally, as did supervisory authors RJ & MCH

## Abstract

Functional near-infrared spectroscopy (fNIRS) promises a non-intrusive way to measure real-time brain activity and build responsive brain-computer interfaces. A primary barrier to realizing this technology's potential has been that observed fNIRS signals vary significantly across human users. Building models that generalize well to never-before-seen users has been difficult; a large amount of subject-specific data has been needed to train effective models. To help overcome this barrier, we introduce the largest open-access dataset of its kind, containing multivariate fNIRS recordings from 68 participants, each with labeled segments indicating four possible mental workload intensity levels. Labels were collected via a controlled setting in which subjects performed standard n-back tasks to induce desired working memory levels. We propose a benchmark analysis of this dataset with a standardized training and evaluation protocol, which allows future researchers to report comparable numbers and fairly assess generalization potential while avoiding any overlap or leakage between train and test data. Using this dataset and benchmark, we show how models trained using abundant fNIRS data from many other participants can effectively classify a new target subject's data, thus reducing calibration and setup time for new subjects. We further show how performance improves as the size of the available dataset grows, while also analyzing error rates across key subpopulations to audit equity concerns. We share our open-access Tufts fNIRS to Mental Workload (fNIRS2MW) dataset[1] and open-source code[2] as a step toward advancing brain computer interfaces.

## 1 Introduction

Functional near-infrared spectroscopy (fNIRS) is a non-invasive sensing technology for measuring brain activity. fNIRS works by shining near-infrared light (650-900 nm) directly onto the brain through the skull and observing changes in the received patterns over time, which reflect changing oxygenation levels in the blood known as the *hemodynamic response*. When regular measurements (2-10 Hz) are collected from several locations over a short window of time (2-60 seconds), the resulting multivariate time series can indicate the intensity of recent mental workload and become a valuable input to a brain-computer interface (BCI). While much work in BCI is aimed at physically challenged users, we focus on "implicit" brain computer interfaces that could help a wider population perform everyday activities at a laptop or desktop. In our intended use case, a user proceeds with mouse and keyboard tasks as usual, while also wearing an fNIRS sensor that is compact and unobtrusive enough to attach to the forehead via a removable strap. The BCI system can then consume fNIRS

---

[1]https://tufts-hci-lab.github.io/code_and_datasets/fNIRS2MW.html [License: CC-BY-4.0]
[2]https://github.com/tufts-ml/fNIRS-mental-workload-classifiers [License: MIT]

input passively, without conscious effort from the user. The fNIRS input drives subtle changes in the user interface or task tailored to the user's moment-to-moment measured mental state. We have demonstrated successful prototype systems (Solovey et al., 2015; Bosworth et al., 2019).

In this work, we consider the problem of developing an effective classifier of mental workload intensity given a short window of fNIRS time-series data. While several efforts have pursued this task before for fNIRS (Coffey et al., 2012; Herff et al., 2014; Aghajani et al., 2017) as well as other BCI sensing technologies (Yin and Zhang, 2017; Saadati et al., 2020), the challenge of building a classifier that can *generalize* to new users remains difficult. Addressing this generalization problem would reduce the required effort to setup a new subject, which would be valuable in a practical BCI setting. Three barriers stand in the way of effective generalization: a lack of large open-access datasets, a lack of standardized protocols following best practices for evaluation, and high variability across subjects.

One common barrier to effective fNIRS-based BCIs is the lack of available data. Previous work typically collects proprietary datasets from only 10-30 subjects. Even in a paid research study, collecting more than an hour of data per user can be difficult because the sensor can eventually become uncomfortable. Most studies do not share data due to the logistical difficulties of human subjects research. While a few open-access datasets for fNIRS data exist (see Table 1), none has more than 30 subjects. Furthermore, the demographic composition of existing data is not always accessible and (if reported) often homogeneous, complicating the goal of BCI for a diverse range of people. Using homogeneous existing datasets to train models is particularly concerning given that NIRS sensor performance may be sensitive to the user's skin color (Wassenaar and Van den Brand, 2005; Couch et al., 2015) as well as dark hair (Chen et al., 2020). This might lead to poor performance for some users, reminiscent of racial disparities observed in face recognition (Buolamwini and Gebru, 2018) and disease detection from pulse oximetry (Sjoding et al., 2020). Improving the diversity and auditability (Raji et al., 2020) of open data is crucial to achieving BCI that works for many people.

Another barrier to progress is the lack of a *standardized* evaluation protocol. Without standardized protocols, different papers may not follow the very same experimental design, making results incomparable and preventing scientific progress. While much is known about best practices for hyperparameter selection and heldout performance estimation in the time series context (Racine, 2000; Mozetič et al., 2018; Cerqueira et al., 2020), without a standard protocol later work may not follow these practices. For example, when evaluating models meant to generalize across subjects, performance should only be reported using never-before-seen subjects. To evaluate a model trained on data from one session and intended for later deployment in that session, it is critical that any heldout data is *temporally distinct* from training data to avoid leakage. Some recent ML-for-BCI works (Mandal et al., 2020; Saadati et al., 2020) report high accuracy on an open dataset (Shin et al., 2018), but do not even reproducibly describe how data was split into training and test sets.

The toughest barrier of all to developing an accurate mental workload classifier is the high *variation* in fNIRS data, which makes generalizing to a new subject or session challenging. Cross-subject differences arise due the physiology of the hair, skin, and skull or differences in brain structure. Across sessions, differences in sensor placement matter. Even within the same session, head motion, body motion, or noise due to the light source or electronics can cause difficulties (Yücel et al., 2021). Given these problems, most studies develop subject-specific classifiers (Herff et al., 2014; Chiarelli et al., 2018; Saadati et al., 2020). Some have tried to develop generic classifiers (Benerradi et al., 2019), while others pursue a transfer-learning approach (Kostas and Rudzicz, 2020). All these different paradigms are hard to compare without open data and benchmarks.

This work attempts to make progress toward overcoming all three barriers. Our contributions are:

1. **We release a large open-access dataset of *68 participants* with a *deployable* fNIRS sensor**. Each subject contributes over 21 minutes of fNIRS recordings from a controlled experimental setting with corresponding labels of workload intensity. In terms of subjects, our dataset is the largest known to us by a factor of 2.5 (see Table 1). We use a slim wearable headband sensor (rather than a more obstructive fabric cap) that does not require precise alignment to landmarks. Furthermore, our sensor is placed directly on the forehead, avoiding most interference due to hair.

2. **We provide a *standardized protocol* for evaluating classifiers and report *benchmark* results on our data under three paradigms of training: subject-specific, generic, and generic + fine-tuning.** We provide instructions and *code* for dividing data into training and test sets to fairly assess generalization potential. Reproducible protocols and benchmarks ensure future studies remain comparable and follow best practices. Our code can be used on other BCI datasets too.

3. **Our large dataset enables fully *generic* classifiers that do not need any training data from a target subject to deliver competitive relative accuracy**. Our generic pipelines trained on data from 64 *other* subjects produce better test accuracies than methods that learn from (limited) target subject data (Fig. 3). The usual need for subject-specific calibration is a nuisance in BCI. The ability to avoid burdensome subject-specific training could help make fNIRS more useable.

4. **The rich demographic information available for each subject in our dataset enables auditing fNIRS classifier accuracy across key subpopulations, especially race.** Preliminary results in Table 2 suggest that homogeneous BCI training datasets may lead to problems when generalizing: models trained only on white subjects classify white subjects somewhat better than other races. While more formal and conclusive analyses are needed, we hope to raise awareness of the importance of such audits in order to develop BCI technologies that work for everyone.

Our previous publication (Wang et al., 2021) pursued some of these goals on a smaller initial dataset of 15 participants. This paper now presents and analyzes data from 68 participants, audits performance across key subpopulations, and assesses generic and fine-tuning paradigms more rigorously.

## 2 Background

**fNIRS background.** For comprehensive reviews of fNIRS as a sensing technology, see Quaresima and Ferrari (2019) and Fantini et al. (2018). Best practices for fNIRS studies are described by Yücel et al. (2021). For coverage of statistical signal processing issues, see Tak and Ye (2014).

**fNIRS as a preferred sensor modality for everyday BCI.** Several physiological data sources can indicate cognitive workload, such as electroencephalograph (EEG) (Berka et al., 2007; Murata, 2005; So et al., 2017), functional magnetic resonance imaging (fMRI) (Michael et al., 2001; Hasegawa et al., 2002), galvanic skin response (GSR) (Shi et al., 2007), and eye tracking (Marshall, 2002; Buettner, 2013). fNIRS has advantages that make it a promising technology for our goal of deployable BCI in natural, non-laboratory settings. fNIRS sensors are removable and portable (unlike fMRI), while providing higher-fidelity measurements of mental activity than GSR or eye-tracking. While EEG is also portable and non-invasive, fNIRS has a finer spatial resolution than EEG (Al-Shargie et al., 2016; Serwadda et al., 2015) and is robust to motion artifacts (Solovey et al., 2009). Several studies have favored using fNIRS and EEG jointly (Shin et al., 2018; Mandal et al., 2020). For simplicity and ease of deployment, our focus is on fNIRS alone. Our fNIRS sensors do not require measurement or landmarking when attached, can use a headband rather than a more intrusive full-head cap, and do not require conductive paste between skin and sensor (unlike many EEG devices).

**Mental workload & n-back experiments.** We wish to obtain labeled intervals of fNIRS data streams in which a participant is reliably exhibiting a known intensity level of mental workload, specifically "working memory workload" (Norman, 2013). The *n-back* task is a standard experimental task for inducing working memory workload (Meidenbauer et al., 2021) whose reliability is supported by meta-analysis (von Janczewski et al., 2021). In an n-back task, the participant is first assigned a difficulty level (a value of $n$, usually 0, 1, 2, or 3). Then, the participant is presented a sequence of stimuli (usually visual images of digits such as "7" or "4"), each one lasting a fixed, brief duration. At each stimulus, the participant indicates if the current symbol is the same as the symbol shown $n$ steps previously or not. For the special case of 0-back, the participant is instructed to indicate an affirmative match for *every* stimulus. The range of $n$ values can simulate differences in workload similarly to real tasks of varying difficulty (Fridman et al., 2018). By recording fNIRS data throughout several n-back tasks of varying $n$, we can obtain *labeled* segments of fNIRS data, using the known difficulty level $n$ as a proxy for each segment's mental workload intensity.

**Classification task.** Our goal is to build a machine learning classifier that, once trained, can consume a fixed-duration multivariate time series of fNIRS data and predict the intensity level of the user's mental workload during that time. Our experiments focus on a binary task distinguishing low workload (0-back) from high workload (2-back). Our dataset also supports fine-grained prediction tasks that classify 4 levels: $n \in \{0, 1, 2, 3\}$. In our intended deployment scenario, we imagine the classifier making predictions every few seconds using the latest fNIRS data as the user performs tasks in real-time. Accurate real-time predictions would make interactive BCI possible.

## 3 Related Work

We now review existing open fNIRS datasets plus methods for mental workload classification.

| Tasks | Dataset & Citation | N | Expt. Duration | NIRS Sensor Info. | Age? | Gender? | Race? | Protocol? |
|---|---|---|---|---|---|---|---|---|
| MW: n-back | Tufts fNIRS2MW (this paper) | 68 | 60 min. | 5.2 Hz; $D = 8$; FD $\{\Delta HbO, \Delta HbR\} \times$ [intensity, phase] 2 locations on forehead via headband | ✓ | ✓ | ✓ | ✓ |
| MW: n-back + other tasks | TU-Berlin Shin et al. (2018)[a] | 26 | 180 min. | 10.4 Hz; $D = 72$; CW $\{\Delta HbO, \Delta HbR\} \times$ [intensity] 36 spatial locations via fabric skull cap | ✓ | ✓ | - | - |
| MW: n-back | Univ. of Oklahoma Mukli et al. (2021)[b] | 14 | 11 min. | 3.9Hz; CW; D=96 $\{\Delta HbO, \Delta HbR\} \times$ [intensity] 48 locations, via skull cap | - | - | - | - |
| MW: driving | Gwangju Inst. of S. & T. Ahn et al. (2016)[c] | 11 | 60 min. | 10 Hz; $D = 16$; CW $\{\Delta HbO, \Delta HbR\} \times$ [intensity] 8 locations, attached to forehead | ✓ | ✓ | - | - |
| MW: arithmetic | TU Graz Bauernfeind et al. (2011)[d] | 8 | 15−20 min | 10 Hz; $D = 156$; CW $\{\Delta HbO, \Delta HbR, \Delta HbT\} \times$ [intensity] 52 locations, via skull cap | - | - | - | - |
| MI: finger / foot tap | Korea U. Bak et al. (2019)[e] | 30 | 13 min. | 10 Hz; $D = 16$; CW $\{\Delta HbO, \Delta HbR\} \times$ [intensity] 8 locations, attached to forehead | - | - | - | - |
| MI: arm raise | U. of Houston Buccino et al. (2016)[f,g] | 15 | 60 min. | 10.42 Hz; $D = 68$; CW $\{\Delta HbO, \Delta HbR\} \times$ [intensity] 34 locations on motor cortex | - | - | - | - |

[a]: http://doc.ml.tu-berlin.de/simultaneous_EEG_NIRS/
[b]: https://www.physionet.org/content/mental-fnirs/1.0/
[c]: http://deepbci.korea.ac.kr/opensource/opendb/
[d]: http://bnci-horizon-2020.eu/database/data-sets
[e]: http://deepbci.korea.ac.kr/opensource/opendb/
[f]: http://dx.doi.org/10.6084/m9.figshare.1619640
[g]: http://dx.doi.org/10.6084/m9.figshare.1619640

Table 1: Comparison of open-access fNIRS datasets known and available as of this writing (Aug. 2021). *Task categories:* MW = Mental workload, MI = motor imagery. *NIRS Sensor Info:* $D$ indicates dimension of observations at each timestep, $D = \#$locations $\times \#$measurements $\times \#$optical data types (intensity or phase). FD = frequency domain; CW = continuous wave. $\Delta HbO$ : oxygenated hemoglobin concentration; $\Delta HbR$ : de-oxygenated hemoglobin conc.; $\Delta HbT$ : total hemoglobin conc. *Age/Gender/Race:* We indicate whether released data contains the age/gender/race of each participant (enabling audits of accuracy by subpopulation). *Protocol:* Whether a standard, reproducible protocol for evaluating MW classifiers exists for the dataset.

## 3.1 Open-Access Datasets for fNIRS

We compare the open-access fNIRS datasets that are known to us and used for building predictive models in Table 1. We assess each one for size, sensor design, auditability, and the availability of standardized evaluation protocols. Some focus on motor imagery (MI) tasks, motivated by the potential of active BCI to improve mobility for physically-impaired individuals. Tasks related to mental workload (MW) are more aligned with our goals of passive BCI for improving everyday BCI interactions. The 26-subject dataset from Shin et al. (2018) is the most similar to ours. In addition to collecting data from 2.5 times more subjects, our dataset has several other advantages: we enforce per-subject quality controls to obtain reliable $n$-back intensity levels; we collect and release more subject-level demographic data to enable audits of subpopulations; and we provide a standard evaluation protocol for several ML training paradigms. Other fNIRS datasets include work by von Lühmann et al. (2020) and Li et al. (2020), but these are less relevant to our study because neither pursues any supervised machine learning tasks or varies working memory intensity.

## 3.2 Classification Pipelines

Existing methods for mental workload classification differ primarily along two axes: whether they use hand-designed or learned feature representations, and whether they are trained using a single source of data (only target subject or only other subjects) or learn from multiple data sources jointly.

**Hand-designed vs. learned representations.** A common approach to workload classification is a two-stage featurize-then-classify pipeline (Coffey et al., 2012; Herff et al., 2014; Aghajani et al., 2017). The first stage consumes each univariate time series and transforms it to a fixed-size feature vector via manually-designed summary functions (popular choices include the mean, variance, minimum, maximum, and slope); the second stage uses a standard classifier on this representation. While simple to implement, these representations may have limited capacity. In contrast, recent BCI efforts apply deep learning approaches with feed-forward (Chiarelli et al., 2018) or convolutional networks (Schirrmeister et al., 2017; Lawhern et al., 2018; Saadati et al., 2020). While more flexible, these methods are vulnerable to overfitting and require more time and memory resources.

**Cross-subject pipelines using multiple data sources.** Some recent efforts in the BCI literature have tried to improve cross-subject generalization when training on data from multiple subjects.

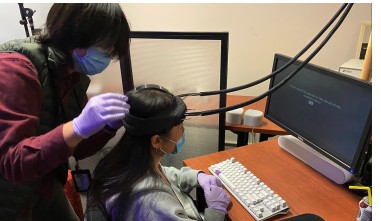

(a) A subject wearing the fNIRS headband, placed by the operator before each experiment began.

| | Asian | White | Latin | Black | Pac. Isl. | other |
|---|---|---|---|---|---|---|
| race | 32 | 27 | 3 | 2 | 1 | 3 |
| | M | F | other | decline | | |
| gender | 26 | 39 | 2 | 1 | | |
| | right | left | unk. | | | |
| handedness | 64 | 3 | 1 | | | |
| | min. | max. | mean | std | | |
| age | 18.0 | 44.0 | 21.71 | 4.01 | | |
| sleep last night (hr.) | 3.5 | 10.0 | 7.29 | 1.21 | | |

(b) Demographics of our eligible cohort (n=68).

Figure 1: Example subject wearing the headband (left) and statistical summary of our eligible cohort (right).

This line of work includes approaches based on contrastive learning (Cheng et al., 2020), domain generalization (Han and Jeong, 2020), optimal transport (Lyu et al., 2021), Riemannian geometric alignment (Congedo et al., 2017), or encoding desired invariances (Kostas and Rudzicz, 2020).

## 4 Dataset

### 4.1 Recruitment and Responsible Data Collection

Volunteer participants were recruited from around the Tufts University campus in Medford, MA from January - July, 2021, using email and posted flyer advertisements. Subjects must be at least 18 years old, speak English, and not have a history of seizures/epilepsy to participate. Procedures were approved by our institution's IRB, and our deidentified dataset was approved for public release (STUDY00000959). Each participant gave informed written consent and was compensated $20 US.

### 4.2 Data Collection Process

We designed an automated framework for collecting high-quality fNIRS data during a controlled n-back experimental setting. Our code implementation is available for others to reuse.

**Pre-experiment.** Each participant completed a survey to collect demographics and contextual information (e.g. sleep habits) to ensure data quality. They were then trained to perform $n-$back tasks. An operator ensured the fNIRS sensor probe was properly placed on the forehead (Fig. 1a).

**Experiment.** Each subject completed 16 blocks of n-back trials; each block contained 40 consecutive trials at the same value of $n$, fixed to 0, 1, 2, or 3. Across the 16 blocks, the order of the values of $n$ was counter-balanced and the same for every subject (see Fig. 2). Each block's 40 trials lasted 80 seconds total, followed by 10-30 seconds of rest. Each trial lasted 2 seconds: a digit between 0 and 9 was shown on the screen for 0.5 seconds, and then hidden for 1.5 seconds. The subject needed to press the left arrow key if the stimuli matched the remembered target and the right arrow key otherwise. Full experimental details are in Appendix C.3.

**Available data.** Our released dataset includes all fNIRS recordings as well as all supplementary data we collected (demographic and contextual information, $n-$back task performance, subjective workload, experiment log, post-experiment interview, etc). Further details are in Appendix C.4.

**Data exclusions.** To ensure quality and consistency, we used several criteria to identify which subjects' data are suitable for classifier evaluation. We selected 68 participants as eligible, out of the total of 87 participants who completed our full experimental protocol. The remaining 19 subjects' data were excluded as follows: 5 subjects' data were excluded because their performance at digit recognition during the n-back tasks was too low (4) or high (1), indicating that their cognitive workload was different from intended. 5 were excluded because their fNIRS sensor settings were different in the early stages of the project. 9 were excluded due to abnormal oxygen measurement dynamics, likely caused by light source leakage. None of our exclusion criteria depended on any machine learning results, but only on factors observed in the raw fNIRS recordings or the subjects' behavior. Data for all subjects, both included and excluded, are available in our public release.

**Eligible cohort summary.** Tab. 1b shows a statistical summary of the eligible cohort. This cohort broadly reflects the Tufts student population in terms of age, racial and gender makeup. The majority of the subjects were of white or Asian race; 6 were from under-represented minorities (URM; 3 Hispanic/Latinx, 2 Black, 1 Pacific Islander), 3 were "other" (mixed race, declined or unknown).

### 4.3 fNIRS Measurements and Preprocessing

We used an Imagent *frequency-domain* (FD) NIRS instrument manufactured by ISS (Champaign, IL, USA). Two custom probes (right and left) were secured to the subject's forehead via a removable headband (Fig. 1a, C.1). The probe used a recent linear symmetric *dual-slope* (DS) sensor design (Blaney et al., 2020a), which beneficially suppressed superficial hemodynamics, instrumental drifts, and motion artifacts (Blaney et al., 2020b). Raw FD-NIRS measurements contain temporal traces of alternating-current intensity ($I$; units: $1/mm^2$) and changes in phase ($\phi$; units: rad) at two wavelengths (690 and 830 nm), using a 110 MHz modulation frequency. From these measurements, we can recover time traces of dynamic changes in oxyhemoglobin ($\Delta$HbO; units: $\mu$mol/L) and deoxyhemoglobin ($\Delta$HbR; units: $\mu$mol/L) from either intensity or phase (Blaney et al., 2020b). See App. C.5 for details. To remove respiration, heartbeat, and drift artifacts, each univariate time-series was bandpass filtered using a 3rd-order zero-phase Butterworth filter, retaining 0.001-0.2 Hz.

Ultimately, for each subject we have a multivariate fNIRS time-series recorded at 5.2 Hz. At each timestep there are $D = 8$ numerical measurements, combining 2 spatial locations, 2 concentrations (oxygenated and de-oxygenated hemoglobin), and 2 optical data types (intensity and phase).

## 5 Methods for Training and Evaluating Classifiers

For each subject, our data collection produces an fNIRS recording lasting over 20 minutes (task duration). We apply a *sliding-window* approach to extract fixed-duration (2-40 sec.) windows from all blocks labeled as $n$-back tasks. Experiments on our dataset suggest that 30 second windows give the highest accuracy (App. B.1). We extract overlapping windows with a stride of 0.6 seconds; with regular predictions every 0.6 seconds, a future interface could respond in near real-time. Train and test data never contain overlapping or adjacent windows: each $n$-back block is assigned to one split.

Given a window duration and stride, we can define each window via a unique endpoint time. Each endpoint represents a specific moment at which we want to classify the user's mental intensity level. Each endpoint (indexed by $p$) for subject $i$ has a fixed-duration time-series $\mathbf{x}_{1:T}^{i,p} = [\mathbf{x}_1^{i,p}, \mathbf{x}_2^{i,p}, \ldots \mathbf{x}_T^{i,p}]$ of $T$ timesteps from that window, each a vector $\mathbf{x}_t^{i,p} \in \mathbb{R}^D$. The same endpoint also has a workload intensity label, $y^{i,p} \in \{0, 1, 2, 3\}$, given by the known value of $n$ within its block. Given many labeled windows, we have thus defined a standard supervised time-series classification task.

### 5.1 Classifiers for Fixed-Duration Multivariate Time Series Windows

Each classifier we tested consumes an fNIRS window $\mathbf{x}_{1:T}^{(i,p)}$ and predicts a workload intensity level. More network architectures and training details are found in App. D.2-D.5.

**Logistic Regression/Random Forest.** Using a featurize-then-classify approach (Herff et al., 2014; Aghajani et al., 2017), we reduce each multivariate ($D = 8$) window to a fixed-size feature vector. We compute from each channel's time-series four scalar summaries: mean, standard deviation, slope, and intercept. Concatenating yields a vector of size 32 ($= 4 \times 8$). Given a feature vector for each endpoint, we train a standard classifier. We try logistic regression (LR) (Hastie et al., 2017) and random forests (RF) (Breiman, 2001). Simple linear boundary classifiers (like LR) are common in fNIRS (Benerradi et al., 2019; Liu and Ayaz, 2018). RF is more flexible yet still easy to train.

**Deep ConvNet.** The deep convolutional neural net (Deep ConvNet) of Schirrmeister et al. (2017) is a strong modern deep learning approach to BCI classification. This network has been widely used for EEG time-series (Amin et al., 2019; Zang et al., 2021) and is easily adapted to fNIRS.

**EEGNet.** Recently, a modified CNN architecture known as *EEGNet* has shown promise across multiple EEG-based BCIs tasks (Lawhern et al., 2018), requiring fewer parameters than DeepConvNet.

**Hyperparameter selection.** For each classifier above (indexed by $c$), we define a grid of $H_c$ hyperparameter values covering a range of model complexities and optimization settings (see App. D). We train a model for each configuration, then select the one with best accuracy on a validation set. This selected model is applied (without retraining) to the test set. Hyperparameter selection is critical to good generalization, but hyperparameter protocols are rarely reported in the BCI literature.

### 5.2 Paradigms for training and evaluation

We consider three paradigms for training a classifier: subject-specific, generic, and generic + fine-tuning. See Fig. 2 for a diagram of each paradigm, which indicates the data used for *training* (informing model parameters), *validation* (selecting among candidate hyperparameters), and *test*

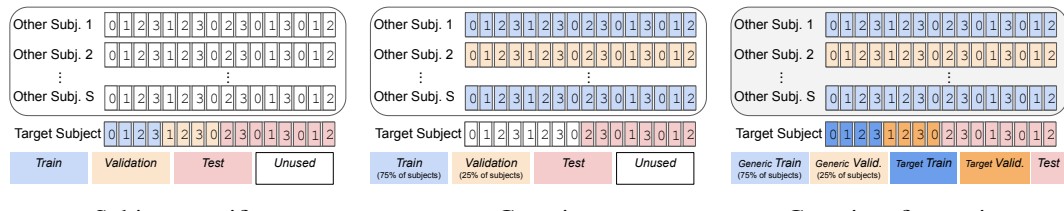

| Subject-specific | Generic | Generic + fine tuning |

Figure 2: Diagrams of how different training paradigms use the available labeled fNIRS recordings in our dataset. Each small box labeled by $n \in \{0, 1, 2, 3\}$ indicates one block of 40 consecutive $n-$back trials (80 sec. of data). From each block, we extract sliding windows and assign all windows to either *Train* (used to update model parameters), *Validation* (used to select hyperparameters), and *Test* (used to evaluate performance on never-before-seen data). Higher $n$ indicates harder mental workload intensity. Our standardized protocol ensures different paradigms are comparable and all heldout accuracy are evaluated on out-of-training-sample data.

(reporting accuracy on heldout data). Our public code release can reproduce these splits. All paradigms are carefully designed to be fairly comparable by using the exact same test set of end points for each target subject (even across different window sizes). Each test set is temporally distinct and separated by a short rest period from any training or validation data. In each paradigm, we select hyperparameters to maximize *validation* accuracy.

**Paradigm 1: Subject-specific.** Given a target subject of interest, the simplest paradigm is to develop a subject-specific model using only the target subject's data (and no data from any other subject). In this paradigm, we train a separate model for each subject and each hyperparameter configuration, yielding $68 \times H_c$ total models for each classifier type $c$ (LR, RF, DeepConvNet, or EEGNet).

**Paradigm 2: Generic.** Our goal with this paradigm is to classify never-before-seen subjects and sessions. Given labeled data from $S$ subjects, we develop a generic classifier that could be applied to data from any target subject. To implement this paradigm with our data, we wish to try generic pools as large as 64 subjects and allow each subject a turn to be in a test set, while training as few models as possible (to minimize complexity). We thus divide all subjects into 17 different buckets of 4 subjects each. From these, we form 17 different development/test splits. Each split uses one bucket of subjects for test; all other buckets combine to form a maximum development set of 64. When the desired pool size $S$ is less than 64, we simply subsample $S$ subjects from the provided 64 at random. This $S$-subject development set is then randomly split by subject into 75% train and 25% validation. We train one model for each split, each generic pool size (we try $S \in \{4, 16, 64\}$), and each hyperparameter configuration, yielding $17 \times 3 \times H_c$ total models for each classifier $c$.

**Paradigm 3: Generic + fine-tuning.** A natural way to leverage *both* sources of development data (from the target subject and from other subjects) is to first pretrain on a large generic pool of $S$ other subjects, and then fine-tune on the target subject, as described for fNIRS data by Wang et al. (2021). To fine-tune to a target subject in our dataset given a pool size of $S$, we begin by taking the best generic model (in terms of validation accuracy) from the train/test split where the target is a test subject. For each hyperparameter configuration, we then initialize the neural network at that checkpoint and execute many iterations of gradient descent training on only the target's training set. This focus on gradient descent limits this paradigm to only our neural network models (DeepConvNet and EEGNet). Thus, we train one separate model for each test subject and each hyperparameter configuration, yielding $68 \times H_c$ total trained models for each deep neural net classifier $c$.

**Domain adaptation with CORAL.** To explore another off-the-shelf way that we might improve target subject performance for non-deep models (LR and RF) using a combination of other subjects and the target, we try correlation alignment (CORAL) (Sun et al., 2015), an unsupervised domain adaptation method showing success in computer vision and in recent domain generalization benchmarks (Koh et al., 2021). In this method, we featurize all windows in both generic and target training sets, apply the CORAL feature transform (learned from the target set) to all generic features, then proceed with model training as usual. Thus, we train one separate model for each subject and each hyperparameter configuration, yielding $68 \times H_c$ total models for each (shallow) classifier $c$.

## 6 Results

To measure the benefits of our new dataset and evaluation protocols, we have focused our experiments on binary classification of low (0-back) from high (2-back) workload intensity (labels are *balanced*:

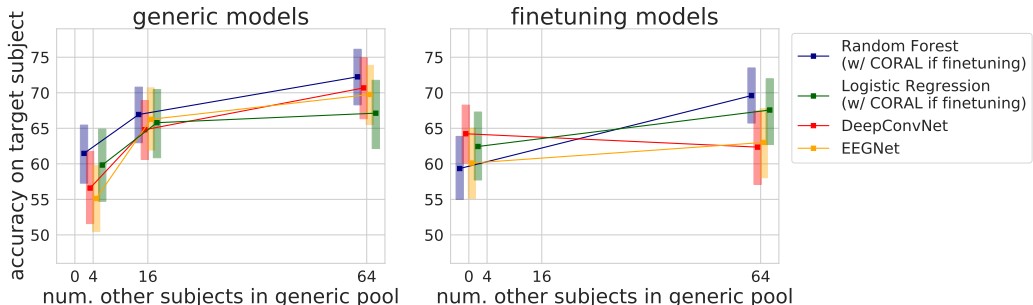

Figure 3: **Classifier accuracy as more subjects are added to the dataset.** The y-axis reports the average test accuracy across all subjects (dark square) plus the 2.5- 97.5 percentile interval of average accuracy across 5000 bootstrap resamples of the test set. *Left:* Models trained under generic paradigm, using a generic pool of 4, 16, and 64 other subjects. *Right:* Models trained under fine-tuning paradigm, considering pool sizes of 64 subjects and 0 subjects (equivalent to subject-specific paradigm). LR and RF use CORAL only for fine-tuning ($S = 64$).

| Train Group | Test Group | RF Accuracy | EEGNet Accuracy |
|---|---|---|---|
| White (21) | White (6) | 77.44 (66.70 - 87.31) | 67.69 (54.11, 80.87) |
| | Asian (6) | 74.18 (63.49 - 85.19) | 64.88 (53.61, 76.54) |
| | URM (6) | 71.43 (60.60 - 82.14) | 62.41 (48.21, 76.31) |
| Asian (26) | Asian (6) | 67.91 (59.01 - 77.56) | 64.14 (53.81, 75.98) |
| | White (6) | 71.29 (59.83 - 81.77) | 64.98 (54.76, 75.01) |
| | URM (6) | 67.39 (57.86 - 77.44) | 65.22 (53.10, 77.43) |

Table 2: **Audit of accuracy across subpopulations by race.** We report accuracies averaged across four train/test splits of subjects. Intervals indicate 2.5th-97.5th percentiles over 5000 bootstrap samples of each test set.

50% accuracy represents a random guess). Our performance metric of interest is test set accuracy, averaged over subjects. To capture uncertainty, we draw 5000 bootstrap resamples of the test set (sampling subjects with replacement, then sampling each included subject's windows with replacement). From each resampled test set, we compute average accuracy, then report 2.5th and 97.5th percentiles across all samples. This bootstrapped interval helps us estimate how the given model's average accuracy might vary across test sets drawn from the same empirical distribution.

In each Finding below, we highlight takeaway messages from our experiments.

**Finding 1: Generic classifiers benefit noticeably from larger training datasets.** In Fig. 3, we show how accuracy improves as the size of the pool of other subjects increases. With only 16 other subjects (roughly similar to many previous studies), we see performance is 66.9% for the best classifier (RF). Using our new larger dataset, we can afford for the first time to assess using a development set of 64 subjects, and we see performance improves further to 72.2%. Collecting even larger datasets could continue to produce noticeable gains.

**Finding 2: On average, generic classifiers outperform both subject-specific and fine-tuning classifiers in our regime of interest where we have only minutes of training data for each of many subjects**. In Fig. 3, we see that when averaging test performance across all subjects, generic+fine-tuning methods do not exceed the performance of purely generic methods on our benchmark. Neither do subject-specific methods (equivalent to fine-tuning when $S = 0$). Of course, only a limited amount of target-specific training data is available (per label $n$, we have 80 sec. for training and 80 sec. validation). While it still seems reasonable that given enough target-specific data, a carefully-designed classifier could improve by fine-tuning, we found this is a challenging task needing further investigation. We provide more discussion of this issue in Sec. B.4.

**Finding 3: Accuracy and preferred paradigm varies considerably across subjects.** In Fig. B.2, we show scatter plots where each dot's location represents the test accuracy of one subject under the two paradigms on y-axis and x-axis. The takeaway here is that considerable variation exists across subjects in terms of overall accuracy (from 30% to 100%) as well as which paradigm is preferred. Despite generic models being better on average, 20 subjects out of 68 show subject-specific accuracy at least 5 percentage points better than the generic model.

**Finding 4: Models trained on homogeneous fNIRS training sets may not generalize as well to other racial groups.** Our large dataset with demographic labels enables audits of how classifiers

generalize across subgroups by race or gender. In Tab. 2, we show what happens when we train a generic classifier on data from one racial group, then test on subjects from each available race (experimental details in App. B.3). When trained on data from white subjects, average test accuracy is highest for white subjects. Compared to the white test group, accuracy drops by at least 2.5 percentage points when tested on Asians, and by at least 5.2 percentage points when tested on URM subjects (Black, Hispanic/Latinx, or Native American/Alaskan/Pacific Islander), across both RF and EEGNet classifiers. Models trained on Asian subjects show no consistent trends across test groups. This "finding" is hardly conclusive, because we have too few non-white, non-Asian subjects (only 6). It does suggest the value of further investigation across subgroups for BCI applications (and the even larger datasets needed to enable such).

**Finding 5: Larger window sizes (≈30 seconds) seem advantageous.** Fig. B.1 shows the results of a systematic comparison of window size using simpler classifiers (LR and RF). It seems there is good reason to prefer window sizes around 30 seconds (where LR achieves a mean accuracy of 62 (bootstrap IQR 61-64)) over shorter windows of 20 seconds or less (which all have 75th percentiles below 61). We hope this clarifies the variation in recommended window sizes in previous literature under protocols of varying reproducibility. For example, some recent work with neural nets use 1 second (Chiarelli et al., 2018) or 3 second (Saadati et al., 2020) windows, while some featurize-then-classify work recommends 25 seconds (Herff et al., 2014).

# 7 Discussion

Here, we discuss the limitations of our dataset and provide an outlook for how it might inspire research directions to advance the fields of machine learning and brain-computer interface design.

**Limitations.** First, our dataset is limited in whom it represents. Because we draw from a convenience sample at our university, ages are skewed toward typical college students and the racial makeup reflects that our campus community is largely white and Asian. Future work is needed to gather datasets with greater numbers for currently under-represented groups. Second, we have only one session recorded for each subject, which does not allow assessment of cross-session generalization. Third, more work is needed with fNIRS data to verify that using $n$-back intensity labels as a proxy for general cognitive workload will yield useful BCI applications, though our prior work suggests it can (Solovey et al., 2015; Bosworth et al., 2019). Finally, the potential of generic+fine-tuning methods bears further investigation. In Sec. B.4, we discuss in detail two candidate explanations for why generic may have performed best in our experiments: the need for even more subject-specific training data and the challenge of distribution shift over time. It still seems reasonable that given *enough* target-specific data, a carefully-designed classifier could reliably improve by fine-tuning.

**Future Directions in ML.** We hope our dataset can inspire advances to several important research directions in machine learning. First, we hope the data inspires effective methods for calibrating a target subject via *domain adaptation from limited time-series data* (supervised and unsupervised). Second, we hope the release of rich demographic information (age, gender, race, handedness, etc.) for many subjects enables research on time-series classifiers that deliver *equitable or fair performance across groups* by design. Finally, the data could inspire improvements in *structured time-series prediction*: while we have treated each window's prediction as separate, ensuring coherence in predictions over time across neighboring windows would be crucial to a successfully deployed BCI.

**Future Outlook for BCI.** BCI research has mostly focused on severely physically challenged users, where higher cost and difficult initial setup might be acceptable. Our vision of implicit BCI in the future is aimed at a wider population. We used a prototype laboratory-grade fNIRS device here, but small, inexpensive wireless fNIRS devices are on the horizon. The basic components required (light source and sensor) pose no fundamental obstacles to use via a slim wearable headband, unlike fMRI devices. As a portable technology that, unlike other brain sensing modalities, is robust to motion, it has tremendous potential for mobile, on-the-go, in-the-wild use outside of research labs (Pinti et al., 2020). We anticipate advancements in solid-state optics, sensing, low-power electronics, and embedded machine learning will make fNIRS widely and generally available in the near future. However, this will require a way to translate the raw fNIRS data stream into actionable predictions of the user's mental state. Our dataset and methods take a step in that direction by providing more accurate data than previously available and also reducing subject-specific setup effort. We foresee future work toward a form of BCI that uses implicit information measured from the brain as a concurrent passive input to improve the efficiency, effectiveness, smoothness, or intuitiveness of otherwise conventional human-computer interaction for a broad range of users.

## Acknowledgments and Disclosure of Funding

This work is supported in part by a grant from Google Inc. and funding from the U.S. National Science Foundation under award HDR-1934553 for the Tufts T-TRIPODS Institute, where CS was a student in the DIAMONDS summer research program advised by MCH. We would also like to thank our colleagues Alex Olwal, Boyang Lyu and Nicholas Mulligan for helpful discussions.

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

# Appendices

## A  Dataset Documentation

### A.1  Quick Links to Data, Code, Datasheet, and Documentation

Our complete dataset, as well as documentation for understanding and using the data, is available at the following URL:

https://tufts-hci-lab.github.io/code_and_datasets/fNIRS2MW.html

Our code for splitting data, training models, and evaluating models according to our "standardized protocol" is available at the following URL

https://github.com/tufts-ml/fNIRS-mental-workload-classifiers

To ensure responsible use of our dataset, we also release a "Datasheet" describing the contents, intended uses, etc. in a standard fashion, following the suggested template of Gebru et al. (2020).

https://github.com/tufts-ml/fNIRS-mental-workload-classifiers/blob/main/
Datasheet-Tufts-fNIRS2MW.pdf

This datasheet will be prominently linked from both the dataset and code websites, so that users will easily find it and learn from it.

These links will be stable and accessible for years to come.

## A.2 License and Responsibility Statement

This dataset is released under a .

We the authors confirm that we bear all responsibility for the content of this dataset, and confirm the license above. We confirm that any data we release at the URLs below is fully deidentified and has the express permission of the person it comes from to be shared openly.

For any questions or further information, please contact the four primary personnel:

1. Zhe Huang
2. Liang Wang
3. Robert Jacob
4. Michael C. Hughes

## A.3 Potential Negative Societal Impacts Statement

fNIRS data collection is non-invasive and does not pose risk of physical or psychological harm to participants. As with any large dataset with deidentified rich metadata, there is a risk of re-identification of some users in the dataset if a bad actor fused the available metadata (e.g. age/race/handedness/etc.) and some external information about the relevant users. We believe this risk is low and potential harms are also low.

Future consumers of this dataset could, against our recommendation and intended scientific uses, try to build classifiers that consume brain activity measurements and predict sensitive attributes (e.g. race or sex) or private medical information (e.g. brain diseases, past drug use, or other conditions). These applications could potentially have more direct harms to some people if realized, but would require additional data far beyond what we have released, as well as physical access to a subject's brain during deployment (difficult to obtain without consent).

## A.4 Hosting and maintenance plan

We will host the dataset files in a shared folder hosted by cloud service provider Box.com. Our official Tufts University accounts give us unlimited storage on this service indefinitely.

We also host a website with documentation and access instructions, as well as a backup copy of the dataset and all code stored at third-party providers (e.g. GitHub). The storage is housed within the Tufts University Computer Science Department. The department server conducts regular backup of the hard drives in which the website is stored. The department has its own dedicated IT team; part of their job is to ensure that all data can be accessed and is backed up. The data, source code, and documentation will be kept on the website and the server indefinitely.

# B Additional Results and Discussion

## B.1 Results: Accuracy vs. Window Size

Below in Fig. B.1, we show for LR and RF classifiers how test-set accuracy (averaged across all subjects) changes with increasing window size from 2 seconds to 40 seconds.

Based on this experiment, we selected 30 seconds as a reasonable window size for all other experiments.

## B.2 Results: How accuracy varies across individual subjects

Below in Fig. B.2, we show how each individual subject in our eligible cohort performed across all 3 paradigms. From this figure, we can see that substantial variations in overall accuracy exist (some subjects are near chance at 50% for this balanced binary classification task, some have test accuracy near 100%). Additionally, while most subjects fair better with the generic training paradigm, many subjects report better accuracy using the subject-specific paradigm.

## B.3 Methods and Results: Audits across subpopulations

Among our 68 eligible subjects, we have data from 27 White subjects, 32 Asian subjects and 6 subjects from under-represented minorities (URM) at our institution. The 6 URM participants include 3 Black, 2 Hispanic/Latinx, and 1 Pacific Islander. For simplicity, we did not include the 3 subjects of "other" race in this analysis.

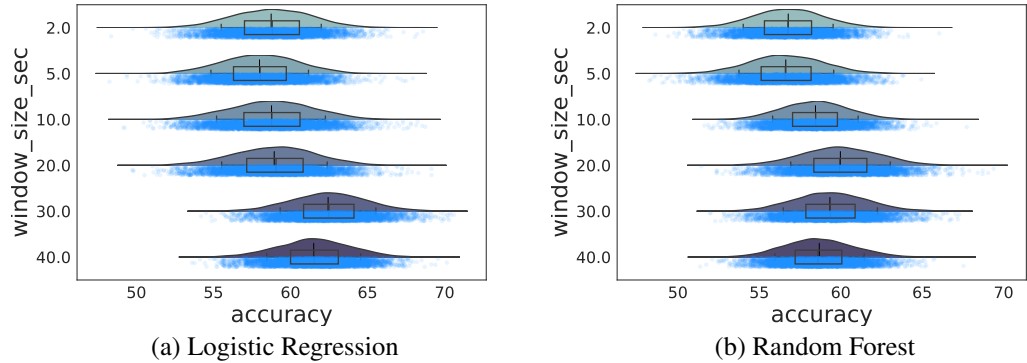

(a) Logistic Regression          (b) Random Forest

Figure B.1: **Test accuracy as a function of window size**, for Logistic Regression (left) and Random Forest (right) classifiers. We indicate mean accuracy on the test set (averaged over subjects) with the tall vertical line. We indicate uncertainty in this value by showing the distribution over 5000 bootstrap samples. Each box plot summarizes this distribution's 25, 50, and 75th percentiles, whiskers indicate 10th and 90th percentiles.

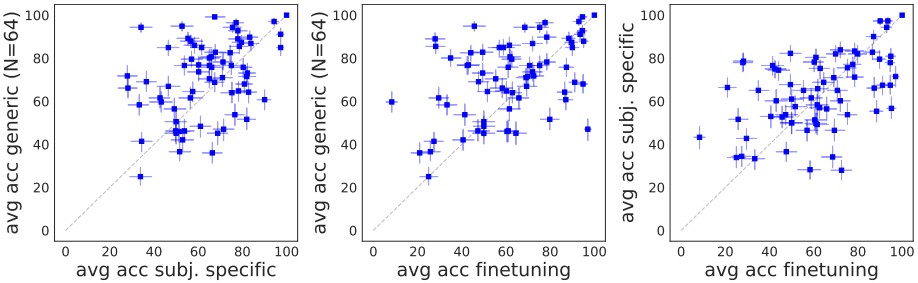

Figure B.2: **Comparison of per-subject test accuracy across paradigms for all 68 subjects in our fNIRS2MW dataset.** Each scatter plot shows one dot per subject. Dots located above the dashed line indicate the y-axis paradigm has higher accuracy; below the dashed line indicates the x-axis paradigm has higher accuracy. Horizontal and vertical lines indicate 2.5%-97.5% intervals of 1000 bootstrap samples of that subject's test set.

To assess how models trained on one homogeneous racial group generalize to other subgroups, we formed a development set from one of the larger racial groups (white or Asian), trained on model using this development set, and then tested that same model on equal sized test sets of 6 subjects each from all three groups (white, Asian, and URM). All 6 URM subjects comprise the test set for the URM group. To form the test set for the white or Asian group, we randomly sampled 6 subjects, leaving the remainder aside (21 white subjects, 26 Asian subjects) as a development set. To avoid results being too sensitive to one random split, we repeated our analysis over four different development/test splits using both Asian and white subjects as a development set.

To train a classifier given each development set, we followed the "generic" training paradigm. Each development set was further split by subject into approximately 75% train and 25% validation. We train on the train set and select model hyperparameters according to the validation set. For simplicity, for this experiment we examined only the best-performing shallow classifier (RF) and the best-performing deep classifier (EEGNet).

### B.4 Discussion: Why doesn't fine-tuning outperform generic?

It was surprising to us that the generic+fine-tuning paradigm did not outperform a generic classifier, which has no chance to specialize to the target subject's data stream. There could be multiple reasons. Below, we provide detailed discussion of the two leading candidate explanations in our view.

**Explanation 1: More training data is needed.** Also, it is possible that given an already 'well-trained' generic neural network model, we need more of a subject's data for fine-tuning to show effect. Note that when fine-tuning (as in all paradigms), we used only the first 25% of the subject's data for training (320 seconds), leaving the second 25% for validation and the last 50% for test.

**Explanation 2: Distribution shift.** The changing nature of fNIRS data from one individual over time, known as *distribution shift* is one possible reason that fine-tuning does not perform better than a

generic paradigm in our results. Our pipeline splits each fNIRS recording in a temporally consistent way: training data comes first, then validation data, then test data. Our training and hyperparameter selection use only the first 50% of data, and are expected to generalize to the second 50% of the data. It is possible that the model that performs best on a subject's validation data does not generalize well to the same subject's test data, due to distribution shift over time (shifts in subject behavior, shifts in sensor placement, sensor failures over time, etc). Despite our best efforts to employ reasonable preprocessing to minimize sensor issues or exclude subjects with poor data quality issues, some minor issues still occur (e.g., some suject's recording shows drift in the raw measurements in the last part of the recording in some channels).

## C  Details of Data Collection

### C.1  Responsible data collection: IRB and COVID-19 safety approval

Because collection occurred during the COVID-19 pandemic in early 2021, we also got approval from the Tufts Integrative Safety Committee (ISC). We followed required sanitation and social distancing practices: experimenters used personal protective equipment and disinfected the fNIRS probe for each subject.

### C.2  Sensor and headband description

The fNIRS sensor and headband used for data collection are visualized in Fig. C.1. The two custom probes (right and left) were secured to the subject's forehead via a removable headband, supervised by an operator from our study team. When possible, the operator avoided any hair between the sensor and the forehead. Occasionally, for some subjects with smaller foreheads and longer hair this was difficult and some hair between the headband and forehead was unavoidable.

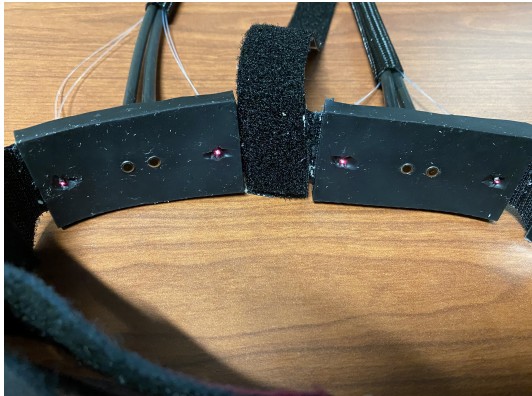

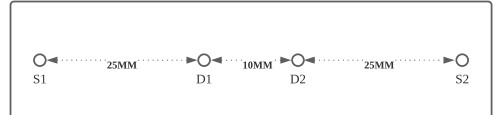

Figure C.1: *Left:* Schematic of functional near-infrared spectroscopy probe. Sources labeled S; detectors labeled D. *Right:* Picture of wearable headband used by each participant, showing the two optical probes. This band is placed on the user's forehead.

### C.3  Task flow description

The task flow each participant went through is in Fig. C.2.

**Instructions.**   Participants were introduced to the $n$-back task via a tutorial video. To minimize artifacts, the video instructed the subject to remain seated, not to talk, and to refrain from adjusting the fNIRS sensors throughout the experiment. After the video, the operator placed the sensor device on the subject.

**Overview of $n$-back design.**   Each subject then completed 16 blocks of n-back trials; each block contained 40 consecutive trials at the same value of $n$, fixed to 0, 1, 2, or 3. Across the 16 blocks, the order of the values of $n$ was predetermined, counter-balanced, and the same for every subject (see Fig. 2). We used a Latin square design (Fig. C.3): a flattened version of a 4 × 4 array filled with 4 different n-back tasks (0, 1, 2, and 3), each occurring exactly once in each row and exactly once in each column.

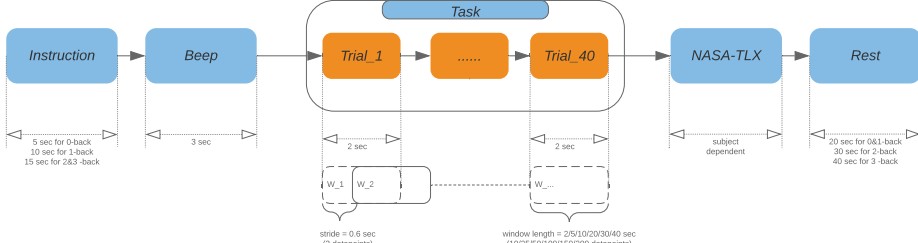

Figure C.2: For each task, subjects underwent pre-task, task, and post-task stages. The duration of instruction and rest period varied among different n-back tasks. Our open source dataset contains raw data from the whole process. In this study, only the task periods (trials 1-40) were pre-processed, analyzed and used to generate the sliding window data.

| 0, 1, 2, 3 | → | 1, 2, 3, 0 | → | 2, 3, 0, 1 | → | 3, 0, 1, 2 |

Figure C.3: All subjects completed 16 $n$-back tasks; each value of $n$ was determined according to this order.

**Each $n$-back block of trials.** The start of each block was indicated with 3 seconds of beeps. Each trial lasted 2 seconds total: a digit between 0 and 9 was shown on the screen for 1.5 seconds, and then hidden for 0.5 seconds. The subject needed to press the left arrow key if the digit was a target (i.e. the same as the digit flashed $n$ trials previously) and the right arrow key otherwise. The system advanced to the next digit trial after 2 seconds regardless of the subjects response (or lack thereof). Completing a single $n-$back block of 40 trials required 80 total seconds. Before each block, participants were shown an infographic to remind them of the current value of $n$ and the desired response process (*left arrow* if current symbol is the same as that $n$-back, and *right arrow* if different).

After each 0 and 1-back block, the subject rested for 20 seconds; 2-back and 3-back blocks had longer rest periods (30 and 40 sec., respectively) in order to improve recovery from the assumed additional difficulty. Once the rest period concluded, the next task among the 16 total tasks immediately began.

**Post-experiment.** At the conclusion of all 16 blocks, participants completed a post-task survey reporting subjective workload via a standard NASA Task Load Index (TLX) questionnaire (Hart, 2006). To further verify quality, we conducted a short interview about the subject's experience, asking 11 open-ended questions. These were later transcribed into text.

### C.4 Available metadata

Our dataset release contains more than just the fNIRS recordings and corresponding workload labels. Below, we describe the additional data available in our release, which can help dataset consumers understand who is in the data and what quality controls are in place.

**Demographic and contextual information** Through the testing interface, we collected demographic data (gender, sex, age, ethnicity, handedness, vision, native language) from the subject. We also collected contextual data (previous head injuries, sleep habits, caffeine intake, drug use, prior experience with biological sensors or $n$-back tasks).

**Cognitive task performance** We measured the subject's performance at the n-back task based on the accuracy of the subject's response for each digit. During our pilot study, subjects were asked to press the space bar for targets. However, we found that when subjects were unsure whether a digit was a target, particularly as the value of $n$ increased, they tended to skip it. Their non-response was recorded identically to an intentional response for a non-target number. In our updated experiment design, we incorporated both arrow keys in order to differentiate between intentional responses for non-targets and non-responses; pressing the correct arrow key for a digit was considered a correct response, while pressing the wrong arrow key or pressing no arrow key were considered incorrect responses.

**Subjective workload** We used the NASA-TLX as a measure of subjective workload. After each $n$-back task, subjects rated each dimension of workload in the NASA-TLX on a 21 point scale.

**Experiment log**    The operator on duty logged any issues that happened during the experiment. This is an essential step which has been neglected by previous public biosensor and fNIRS datasets. fNIRS sensors are very sensitive to the environment and can be polluted by many factors. We require the operator to log issues such as interference from a subject's hair and light leaking (often occurring when the shape of a subject's forehead does not match the curve of the headband). We also ask the operator to report any DC intensity detector over-voltage warnings. The higher DC intensity we set, the better data we are able to collect; however, we want to avoid over-voltage and saturation, which may cause the fNIRS system to shut down automatically.

**Post-experiment interview**    We asked 11 open-ended questions in a recorded interview. The questions targeted the subject's physical comfort, emotions, and experience with the experimental tasks, testing interface, and hardware.

### C.5    Dual-slope processing of fNIRS data

From intensity ($I$) and phase ($\phi$) measurements, we can recover time traces of the dynamic changes in oxyhemoglobin ($\Delta$HbO) and deoxyhemoglobin ($\Delta$HbR), by calculating the spatial dependence (slope versus source-detector distance) of said optical measurements. We used a source-detector arrangement known as a linear symmetric *dual-slope* (DS) set (Fantini et al., 2019), as shown in Fig. C.1. This probe and arrangement design, previously published by Blaney et al. (2020a), beneficially suppresses superficial hemodynamics, instrumental drifts, and motion artifacts (Blaney et al., 2020b).

Following describes conversion of optical data to hemodynamic data with DS, details can be found in Blaney et al. (2020b). A measurement of *single-distance* (SD) $I$ ($SDI(\rho, \lambda) = ln(\rho^2 I(\rho, \lambda))$) or SD $\phi$ ($SD\phi(\rho, \lambda) = \phi(\rho, \lambda)$) will be referred to as $SDY$ for simplicity (where $\rho$ is source-detector distance and $\lambda$ is wavelength). The average slope of $SDY$ versus $\rho$ in the DS set ($DSY$) is calculated:

$$DSY(\lambda) = \frac{1}{2} \left( \frac{SDY_1(\rho_2, \lambda) - SDY_1(\rho_1, \lambda)}{\rho_2 - \rho_1} + \frac{SDY_2(\rho_2, \lambda) - SDY_2(\rho_1, \lambda)}{\rho_2 - \rho_1} \right) \quad (1)$$

where $SDY_i(\rho_j)$ is the $i^{th}$ SD measurement at distnace $\rho_j$. This $DSY(\lambda)$ is measured during a baseline period to yield $DSY_0(\lambda)$. Changes from this baseline ($\Delta DSY(\lambda, t) = DSY(\lambda, t) - DSY_0(\lambda)$) were used to calculate $\Delta\mu_{a,Y}(\lambda, t)$, at time $t$:

$$\Delta\mu_{a,Y}(\lambda, t) = -\frac{\Delta DSY(\lambda, t)}{DSF_Y(\lambda)} \quad (2)$$

where $DSF_Y(\lambda)$ is the differential slope factor which depends on the absolute optical properties of the tissue. For this work an absolute absorption coefficient of $\mu_a = 0.01$ 1/mm and an absolute reduced scattering coefficient of $\mu_s' = 1$ 1/mm were assumed for each wavelength. Finally, $\Delta HbO_Y(t)$ and $\Delta HbR_Y(t)$ were calculated:

$$\begin{bmatrix} \epsilon_{HbO}(\lambda_1) & \epsilon_{HbR}(\lambda_1) \\ \epsilon_{HbO}(\lambda_2) & \epsilon_{HbR}(\lambda_2) \end{bmatrix}^{-1} \begin{bmatrix} \Delta\mu_{a,Y}(\lambda_1, t) \\ \Delta\mu_{a,Y}(\lambda_2, t) \end{bmatrix} = \begin{bmatrix} \Delta HbO_Y(t) \\ \Delta HbR_Y(t) \end{bmatrix} \quad (3)$$

where $\epsilon_C(\lambda_i)$ is the extinction coefficient for chromophore $C$ at wavelength $\lambda_i$. All further analyses utilized these DS-derived changes of chromophore concentrations for both data-types (*i.e.* $HbO_I(t)$, $HbR_I(t)$, $HbO_\phi(t)$, $HbR_\phi(t)$).

## D    Details of Classification Methodology

### D.1    Summary of resources needed to reproduce experiments

The resources needed can be divided into training classifiers and evaluating the predictions of classifiers on new data. The most expensive step in terms of runtime is training, and the most expensive paradigm to use is the generic paradigm (because the training set is so large).

**Training resources needed.**    A summary of the runtime training costs is given in Table D.1 below. The Generic and Generic + Fine-tuning runtimes are for the largest generic pool size scenario which used $S = 64$ subjects.

The most expensive models we tried were those based on deep neural networks. Both DeepConvNet and EEGNet are trained using GPU. We train both the DeepConvNet and EEGNet for 300 epochs for

the Subject-Specific and Fine-tuning experiments. We used 600 epochs when performing Generic experiments, as the size of the training set is much bigger and convergence takes longer.

Among the "shallow" classifiers, LR and RF models are trained using CPU only implementations in scikit-learn (Pedregosa et al., 2011). RF training typically completes within a minute or less across all paradigms, and LR training is typically much faster than the RF.

| Paradigm | Num. Models trained per hyperparam. config | Typical RF Runtime (per model) | Typical Runtime for DeepConvNet (per model) |
|---|---|---|---|
| Subject-Specific | 68 (num. subjects) | <1s | $\sim 6s$ |
| Generic | 17 (num. buckets) | $\sim 60s$ | $\sim 1200s$ |
| Generic + Fine-tuning | 68 (num. subjects) | $\sim 30s$ | $\sim 6s$ |

Table D.1: Resources needed to train models under each paradigm. Note that generic+fine-tuning assumes that generic has already been completed (we don't list the same prerequisite resources twice).

**Evaluation resources needed.** Once we have trained models for every hyperparameter configuration, it is fast (less than 1 minutes) to evaluate each pretrained model on validation data, select the best configuration, and report that model's test accuracy.

### D.2 Logistic Regression : description, hyperparameters, and implementation details

To train LR, we perform penalized empirical risk minimization using the cross-entropy loss function and an L2-penalty on the weight coefficients. Parameters are estimated via the L-BFGS algorithm. For all experiments conducted in ths study, We search 11 possible values for the L2-penalty strength, logarithmically spaced from $10^{-5}$ to $10^5$. All other settings use the defaults in the SciKit-Learn softwarepackage (Pedregosa et al., 2011) Best hyperparameters are chosen according to performance on validation set.

### D.3 Random Forest : description, hyperparameters, and implementation details

We train RF using SciKit-Learn softwarepackage (Pedregosa et al., 2011). For all experiments conducted in this study, we search four values for max_features ([0.166, 0.333, 0.667, 1.0], this defines the fraction of all features to randomly include in each tree) and three values for the min_samples_leaf ([4, 16, 64], this defines the minimum size of a leaf node in the decision tree in terms of number of training examples). All other settings use the defualt from the software. Best hyperparameters are chosen according to performance on validation set.

### D.4 Deep Conv. Net. : description, hyperparameters, and implementation details

As an exemplar of a competitive modern deep learning architecture for BCI applications, we use the Deep ConvNet architecture proposed by Schirrmeister et al. (2017). This deep convolution neural net architecture (DeepConvNet) is designed particularly for general feature extraction from brain signals (especially EEG) and has been widely used and compared against in previous work (Lawhern et al., 2018; Amin et al., 2019; Zang et al., 2021).

**Architecture.** The network consists of four convolution-max-pooling blocks, with the first block performing a temporal convolution followed by a spatial convolution, and the three other blocks performing standard 2D convolutions. Batch normalization (Ioffe and Szegedy, 2015) is used within each block, followed by ELU (Clevert et al., 2015) activation. We re-implemented the network in pytorch (Paszke et al., 2017) closely following the original implementation.

Details can be seen in table below:

We set $F1 = 25$, $F2 = 25$ and the number of output filters for the third, fourth and fifth convolution to be 50, 100 and 200 as in Schirrmeister et al. (2017). We adjust the strides to adapt to our data shape.

**Optimization.** We train the model using Adam optimizer (Kingma and Ba, 2014) as in Schirrmeister et al. (2017). We use full batch gradient descent when training the model.

| Block | Layer | num. filters | kernel size |
|---|---|---|---|
| 1 | Conv2D | F1=25 | (1, 5) |
| | Conv2D | F2=25 | (C=8, 1) |
| | BatchNorm2D | | |
| | ELU | | |
| | MaxPool2D | | (1, 2) |
| | Dropout | | |
| 2 | Conv2D | 50 | (1, 5) |
| | BatchNorm2D | | |
| | ELU | | |
| | MaxPool2D | | (1, 2) |
| | Dropout | | |
| 3 | Conv2D | 100 | (1, 5) |
| | BatchNorm2D | | |
| | ELU | | |
| | MaxPool2D | | (1, 2) |
| | Dropout | | |
| 4 | Conv2D | 200 | (1, 5) |
| | BatchNorm2D | | |
| | ELU | | |
| | MaxPool2D | | (1, 2) |
| | Linear Classification | | |

Table D.2: DeepConvNet architecture, where C = number of channel, F1 = number of temporal filters, F2 = number of spatial filters

**Hyperparameters.** For subject-specific and generic model experiments, we searched the learning rate in [0.001, 0.01, 0.1, 1.0, 10.0] and dropout probability in [0.25, 0.5, 0.75]. For fine tuning experiments, we search learning rate in [0.0001, 0.001, 0.01, 0.1, 1.0] and dropout probability in [0.25, 0.5, 0.75]. Best hyperparameters are chosen according to performance on validation set.

### D.5 EEGNet : description, hyperparameters, and implementation details

EEGNet has shown promising performance across multiple paradigms of EEG-based BCIs (e.g., P300 visual-evoked potential, error-related negativity, movement-related cortical potential and sensory motor rhythm) (Lawhern et al., 2018). Lawhern et al. (2018) showed that EEGNet is able to achieve competitive results compared to DeepConvNet (Schirrmeister et al., 2017) with far fewer parameters. We expect EEGNet to be a potential architecture for fNIRS data, since both the EEG and fNIRS represent relatively fast (>2Hz) time-varying signals that measure the brain activity (electrical or hemodynamic).

**Architecture.** The network contains three convolution blocks, with the first block performing a temporal convolution, followed by the second block performing a depthwise convolution (Chollet, 2017), and the third block performing a separable convolution (Chollet, 2017), which is a depthwise convolution followed by a pointwise convolution. Details can be seen in table below.

We set $F1 = 4, F2 = 8, D = 2$ as in Lawhern et al. (2018). We set the first convolution kernel length to be 3 (half the sampling rate, as recommended in Lawhern et al. (2018)). We set the kernel length of the Separable convolution layer to 3 (instead of 16 in Lawhern et al. (2018))to adapt to our chosen window size. We re-implemented the network in pytorch (Paszke et al., 2017) closely following the original implementation. We didn't try other kernel size, but it is worth noting that the EEGNet is designed originally for EEG signals, so other better choice are possible and we leave it for future research.

**Optimization.** We train the model using Adam optimizer (Kingma and Ba, 2014) as in Lawhern et al. (2018). We use full batch gradient descent when training the model.

| Block | Layer | num. filters | kernel size |
|---|---|---|---|
| 1 | Conv2D | F1=4 | (1,3) |
|   | BatchNorm2D | | |
| 2 | Depthwise Conv2D | D(=2) * F1 | (C=8, 1) |
|   | BatchNorm2D | | |
|   | ELU | | |
|   | AvgPool2D | | (1, 4) |
|   | Dropout | | |
| 3 | Separable Conv2D | F2=8 | (1, 3) |
|   | BatchNorm2D | | |
|   | ELU | | |
|   | AvgPool2D | | (1, 8) |
|   | Dropout | | |
|   | Linear Classification | | |

Table D.3: EEGNet architecture, where C = number of channels, F1 = number of temporal filters, D = depth multiplier, F2 = number of pointwise filter

**Hyperparameters.** For subject-specific and generic model experiments, we searched the learning rate in [0.001, 0.01, 0.1, 1.0, 10.0] and dropout in [0.25, 0.5, 0.75]. For fine tuning experiments, we search learning rate in [0.0001, 0.001, 0.01, 0.1, 1.0] and dropout in [0.25, 0.5, 0.75]. Best hyperparameters are chosen according to performance on validation set.

