# OpenReview forum: "The Tufts fNIRS Mental Workload Dataset & Benchmark for Brain-Computer Interfaces that Generalize"
_NeurIPS.cc/2021/Track/Datasets_and_Benchmarks/Round2 — NeurIPS 2021 Datasets and Benchmarks Track (Round 2)_

### Official Review · Reviewer_hptq · 2021-09-19
**Review for the Tufts fNIRS paper**

**Rating:** 7
**Confidence:** 2

**Strengths:**

The paper offers the largest (n=68) dataset of its kind that is collected in a well-designed study setting. I particularly appreciate the paper delving into the issue of auditability and pointing out that the demographic differences of subjects may influence the effectiveness of the trained system. The experiments run to demonstrate both the evaluation protocol and the effectiveness of the dataset are convincing and lay out useful guidance for future study in this topic. Finally, the paper is easy to follow and clearly explains the gap in the literature that motivated the work.


**Weaknesses:**

While I appreciate the paper’s discussion around auditability and the importance of including those in the marginalized/underrepresented communities, the dataset itself is not collected in ways that would actually mitigate the underrepresentation and the study demographic largely represents the university population. A different sampling procedure may have been useful to curate a dataset that better represents the underrepresented communities. I was also slightly surprised that the generic+fine tuning classifier performed worse than the generic classifier. I could imagine a few different explanations for this, but it could have been helpful if the paper delved into this a little bit more to actually explain the performance difference between the two conditions.

**Additional Feedback:**

.

**Clarity:**

The presentation is clear and easy to follow.


**Correctness:**

The data collection procedure was carefully designed and executed and the experiments are reasonable and provide effective benchmarks. However, given that the amount of training data available varied significantly between the subject-specific and generic condition, the header for Finding 2: On average, generic classifiers outperform both subject-specific classifiers and fine-tuning classifiers may be overclaiming.


**Documentation:**

Documentation is clear and all the relevant data (including those from the excluded participants) is included.


**Ethics:**

The study and the data release were approved by the authors’ institution’s IRB.


**Relation To Prior Work:**

The work captures the gaps in prior work and effectively argues for its contribution in bridging those gaps.


**Summary And Contributions:**

The paper identifies three challenges to developing effective classifiers for mental workload intensity: 1) the lack of publicly available data, 2) a standardized evaluation protocol, and 3) high variation in fNIRS data used to build the classifiers based on the (demographic) characteristics of the subject. The paper aims to address these challenges by releasing a new public fNIRS dataset of 68 participants, by proposing an evaluation protocol that’s generalizable across different subjects, and by providing demographic metadata along with the dataset to enable auditing of the trained systems.

---

> ### Author Response · Authors · 2021-09-27
> **Reply to reviewer hptq: Addressing limitations in how we sampled participants, explaining the generic vs fine-tuning gap**
>
> Dear Reviewer hptq, thank you for your in-depth review and suggestions. We respond to them below:
>
> ## RE "A different sampling procedure may have been useful to curate a dataset that better represents the underrepresented communities"
>
> We agree. We used a convenience sample in this study and thus we did not collect as many samples from underrepresented groups as we could have under alternative strategies. We do mention this in our “Limitations” paragraph under Sec. 7, but we will revise to be more clear. We have plans to pursue a new recruitment and sampling procedure in future work. We hope our work inspires a broader trend toward better representation in future BCI work.
>
> ## RE Explain more about the performance difference between generic and generic+fine tuning
>
> Thank you for the thoughtful comment. We were also surprised by the result too. We will revise the discussion section and supplement to try to address possible causes more thoroughly.
>
> There could be multiple reasons that the generic classifier currently performs better than the fine-tuning approach. Two possible reasons might be:
>
> #### Reason 1: Distribution shift over time
>
> Our pipeline splits each fNIRS recording in a temporally consistent way: training data comes first, then validation data, then test data. Our training and hyperparameter selection use only the first 50% of data, and are expected to generalize to the second 50% of the data. It is possible that the model that performs best on a subject’s validation data does not generalize well to the same subject’s test data, due to distribution shift over time (shifts in subject behavior, shifts in sensor placement, sensor failures over time, etc). Despite our best efforts to employ reasonable preprocessing to minimize sensor issues or exclude subjects with poor data quality issues, some minor issues still occur (e.g. subject 27’s recording shows drift in the raw measurements in the last part of the recording in channels AB_I_O and CD_I_O).
>
> #### Reason 2: More subject-specific data are required for fine-tuning
>
> It is also possible that given an already ‘well-trained’ generic neural network model, we need more of a subject’s data for the fine-tuning to show a noticeable effect. When fine-tuning (as in all paradigms), we used only the first 25% of the subject’s data for training (320 seconds), leaving the second 25% for validation and the last 50% for test. We could examine using more data for the training phase to test this hypothesis.
>
> We think this is a complicated issue, as the two causes above may be interlinked or both true. We plan to investigate this more in future works.
>
>
>
> ## RE Wording of the header for Finding2 may be overclaiming
>
> Thank you for pointing this out, we will soften the language in the revision and be clear that in our analysis regime of interest, the generic classifier has access to far more training data in total than the subject-specific classifier.
>
> We do suggest that the regime in which our claims hold is the one that is potentially most common and useful in practice: after amassing modest amounts of data from many subjects, we would like to build a classifier that can generalize to a brand-new subject without needing more than a few minutes of data from the new subject.

---

### Official Review · Reviewer_6mqk · 2021-09-19
**The authors release the largest fNIRS-to-mental workload dataset, which may enable development of robust and generalizable passive BCI applications in the future. The authors also propose a "benchmark" for evaluating classifier performance on mental workload, although, this is not stated explicitly in the paper.**

**Rating:** 9
**Confidence:** 3

**Strengths:**

This dataset is the largest of its kind, with the next largest fNIRS-to-mental workload dataset being comprised of 26 subjects. The dataset also seems to be carefully curated with subject demographics (i.e. gender, race, handedness) being obtained from a questionnaire for each subject. This sort of information will be important going forward for developing generalizable BCI applications, since the hemodynamic response obtained by by fNIRS depends not only on the stimuli, but also demographic-specific features such as skin color.

**Weaknesses:**

The authors are technically also proposing a “benchmark” in addition to the fNIRS dataset. The benchmark consists of the standardized pre-processing steps (i.e. filtering, sliding window extraction), and training/testing splits for the different tracks (i.e. subject-specific, generic, generic + fine-tuning). I recommend that the authors adjust the wording throughout the text, and in the title to specify that this is also a benchmark.

**Additional Feedback:**

-	In Section 3.2, Cross-subject pipelines, you might also include recent work on contrastive learning to improve cross-subject generalization in biosignal classification: JY Cheng, et al. “Subject-aware contrastive learning for biosignals,” arXiv:2007.04871 [cs.LG]
-	In Figure B.2: For clarity, it would be useful to label the axes as “mean accuracy”


**Clarity:**

The paper is easy to understand, and the information is presented in an organized fashion.

**Correctness:**

It is correctly stated in the title and throughout the article that the authors are releasing an fNIRS dataset consisting of measurements in 64 subjects performing mental workload tasks. However, the authors should clarify in the title, and throughout the text that they are also proposing a “benchmark”.

**Documentation:**

The authors provide two links. The first is a landing page with a link to download the dataset, and in-depth information about the structure of the dataset. Another link goes to a Github page with code that reproduces each of the experiments detailed throughout the text.

**Ethics:**

None that I know of.

**Relation To Prior Work:**

The authors provide a near-comprehensive overview of previous publicly released fNIRS to mental workload datasets. Below is one publicly available dataset that wasn’t mentioned. Although, this dataset is not as large as the one released by the authors.
-	Mukli, et al. “Mental workload during n-back task captured by TransCranial Doppler (TCD) sonography and functional near-infrared spectroscopy (fNIRS) monitoring. PhysioNet. https://doi.org/10.13026/zfb2-1g43.


**Summary And Contributions:**

In this work, the authors release a dataset comprised of functional near-infrared spectroscopy (fNIRS) measurements across 68 subjects during mental workload tasks of varying difficulty. Additionally, the authors outline a “protocol” for evaluating classifier performance on mental workload difficulty on pre-designed splits of the data. Multiple tracks are presented in which the training and testing splits are designed to assess both machine learning- and deep learning-based classifiers models ability to generalize across subjects, and potential for fine-tuning on limited data from the target subject. The dataset along with a detailed description of the data format is linked in the paper, along with code to create the data splits is released on github.

---

> ### Author Response · Authors · 2021-09-27
> **Reply to reviewer 6mqk: Adding 'benchmark' to contributions, discussion of dataset from Mukli et al**
>
> Dear Reviewer 6mqk, thank you for your in-depth review and suggestions. We respond to them below:
>
> ### RE Adjust the wording to mention that the paper is also proposing a ‘benchmark’:
>
> Thanks for your suggestion. We will adjust the wording in the abstract, the listing of contributions within the Introduction, and throughout the text as appropriate to make the fact that we introduce both a dataset and a “benchmark” more clear. We will also consider adjusting the title along the lines of “The Tufts fNIRS to Mental Workload Dataset and Benchmark: …”, if revised titles are allowed by the submission process at this stage.
>
> ### RE Related dataset by Mukli et al
>
> Thank you for bringing the work by Mukli et al. (2021, https://www.physionet.org/content/mental-fnirs/1.0/) to our attention. We will cite and discuss it in our revision, and will include it in our listing of related mental workload (MW) datasets in Table 1. This open-access data could be used to build mental workload classifiers like the ones we study.
>
> This dataset contains 11 minute recordings from 14 subjects across 3 levels of n-back difficulty. We have longer recordings (60 minutes) from more subjects (68). We also introduce a machine learning benchmark; Mukli et al do not try to build any classifier to recognize the mental workload intensity level given a short-duration fNIRS window.
>
> ### RE Related work on cross-subject generalization
>
> Thank you for bringing the work of Cheng et al. (2020, https://arxiv.org/pdf/2007.04871.pdf) to our attention. We will cite it in our discussion of  cross-subject methodology (current Sec. 3.2) as an example of how contrastive learning can be used to attack the cross-subject problem.
>
> ### RE Axes label for Figure B.2.
>
> Thanks for your suggestion. We will revise the label and caption of Figure B.2 to be more clear.

---

### Official Review · Reviewer_GmA5 · 2021-09-21
**Questions about dataset format and related work**

**Rating:** 6
**Confidence:** 3
**Correctness:** .
**Clarity:** .

**Strengths:**

.

**Weaknesses:**

A quick search on the internet pulled up some related work which seems like it should have been included given that the authors mentioned they aimed to include all datasets "known and available as of this writing (Aug. 2021)."

Open Access Multimodal fNIRS Resting State Dataset With and Without Synthetic Hemodynamic Responses
https://www.frontiersin.org/articles/10.3389/fnins.2020.579353/full

Brain correlates of motor complexity during observed and executed actions
https://www.nature.com/articles/s41598-020-67327-5

These both seem to mention the lack of datasets and also discuss workload as an aspect of their datasets. How does your work compare to them?

Also, the format of the dataset doesn't seem to conform to any standard. This is not my field, but it seems like in the database https://openfnirs.org/data/ the Snirf format (Shared Near Infrared Spectroscopy Format) is used for all fNIRS datasets. It is not mentioned in your paper why you didn't use it. Standard formats are very important for exchanging data although they are not all perfect so an explanation about why your format will have more impact than using the snirf will clarify this decision.


**Additional Feedback:**

.

**Documentation:**

.

**Ethics:**

The paper states: "Procedures were approved by our institution’s IRB, and our deidentified dataset was approved for public release (STUDY00000959)"

I do not see any other concerns with this work.

**Relation To Prior Work:**

.

**Summary And Contributions:**

The work presents a fNIRS of working tasks for long durations of time.

---

> ### Author Response · Authors · 2021-09-27
> **Reply to reviewer GmA5: Discussion of 2 requested papers and our chosen CSV format**
>
> Dear Reviewer GmA5,
> Thank you for your in-depth review and suggestions. We respond to them below:
>
> ## RE Comparison to two other related datasets
>
> Thank you for suggesting these works. We are studying them and will cite both in the revised version of our paper.
>
> ### Paper 1: von Lühmann et al. (2020)
>
> The first paper “Open Access Multimodal fNIRS Resting State Dataset With and Without Synthetic Hemodynamic Responses” by von Lühmann et al. (2020), provides two sets of fNIRS data that focus on the brain’s “resting state”. One dataset contains 5 min of resting state data from 14 participants and the other contains 10 min of resting state data from 14 participants. In all cases, participants were recorded sitting still while looking at “at a fixation cross on a black screen”, but not performing any cognitive tasks, which is the focus of our study. No machine learning was performed in that study. We don’t think that the recordings from that study could be used to train a mental workload classifier like ours that can differentiate between "high" and "low" intensity working memory activities.
>
> This dataset is available for download at this URL: https://www.nitrc.org/frs/?group_id=1071, but requires that the user first register for a free account on the NeuroImaging Tools and Resources Collaboratory (NITRC).
>
> ### Paper 2: Li et al. (2020)
>
> The second paper “Brain correlates of motor complexity during observed and executed actions” by Li et al (2020, https://www.nature.com/articles/s41598-020-67327-5.pdf) conducted an experiment where participants watched a video of a subject inserting an object into a box and then used their own hands to place a similar object into a similar box in front of them. The actions shown and performed were either “simple” (insert into box with wide open top) or “complex” (insert into narrow slot on closed box). Given its focus on coordinated hand motion, this study belongs more in the “motor imagery (MI)” category of fNIRS studies that we outline in our Table 1, as it relates to how brain activity changes as subjects think about moving their body or actually move their body. No mental workload (“MW”) task is mentioned or applied in this paper. While the raw data was analyzed to understand if differences can be measured across tasks, it does not seem that any “classification” or other supervised machine learning task was studied. Finally, this study differs from ours in terms of the region of the brain that is measured. Our study focuses on the prefrontal cortex (PFC), which is relevant to working memory. Li et al. instead examined regions more appropriate to motor activity, including the primary motor cortex (M1) and the ventral premotor cortex (PMv). See Li et al’s Figure 3 for a clear diagram that they are looking at different areas of the brain than our study, which focuses on the prefrontal cortex immediately behind the forehead.
>
> The Li et al. dataset is not publicly available for anyone to download. Instead, it is “available from the corresponding author on reasonable request”.
>
>
> ## RE Format of the dataset and why we selected CSV instead of SNIRF
>
> We selected our dataset format -- plain-text comma-separated value (CSV) files -- to be as universal and widely used as possible especially for the broader machine learning community even beyond BCI. Most major programming languages (e.g. Python, Matlab, C++, Java) provide libraries of tools for reading and writing CSV files. CSV files are well-supported by modern machine learning software stacks (e.g. the Python ecosystem of NumPy, Pandas, scikit-learn). CSVs are also easy for a human reader to inspect and understand using any text editing program or spreadsheet software (like Microsoft Excel). As a stable format with a wide user base, CSVs should also be easy to read far into the future, ensuring the long-term usability of our dataset.
>
> Our intended audience is broad, including machine learning (ML) researchers with methodological interests in time series classification as well as researchers who specialize in brain-computer interfaces (BCI). We did not wish to select a format that would be unfamiliar to a more general ML audience that is new to BCI and might already be struggling to understand other BCI concepts involved in our dataset. We also wanted to avoid requiring additional software dependencies. The suggested SNIRF file format would require additional packages to be installed in the user’s environment, potentially leading to installation issues, bugs, or long-term maintenance issues.
>
> That said, we appreciate the SNIRF format’s goals to comprehensively document NIRS data. We will cite this work on our website and consider providing data in SNIRF format in the future, especially if that format becomes more widely used (currently, most datasets in our Table 1 are not yet provided in SNIRF format; for example Shin et al and Bauernfeind et al. both use Matlab “.mat” files available).

---

> > ### Comment · Reviewer_GmA5 · 2021-10-14
> > **> RE Format of the dataset and why we selected CSV instead of SNIRF**
> >
> > Sorry for the delay I am just seeing this reply now.
> >
> > The points made also support sharing images in CSV format.
> >
> > Your position seems to be that users will write their own custom parsers for your CSV data. This makes the data difficult to work with.
> >
> > Your argument that SNIRF format libraries are buggy has the same solution where you could also just expect the user to implement their own SNIRF format parser.
> >
> > I was asking why this format is not a good format. I am not arguing that it is but I believe standardized formats are very important for sharing data.

---

> > > ### Author Response · Authors · 2021-10-15
> > > **RE Format of the dataset and why we selected CSV instead of SNIRF**
> > >
> > > Thanks for your response. We share your overall goals that standard formats are good for the community.
> > >
> > > As part of our code release, our open-source Python code contains routines that users can easily load and use our current CSV-formatted data within common Python ML pipelines (e.g. sklearn or pytorch) without needing to write their own parsers from scratch.
> > > You can find this code in our repository: https://github.com/tufts-ml/fNIRS-mental-workload-classifiers
> > > We are actively improving both the code, examples, and documentation. We welcome issues and pull requests.
> > >
> > > We actively hope that our suggested CSV format (which integrates well with our code) might become one "standard" way to release fNIRS data with annotations intended for use in building predictive models. We hope to someday soon release preprocessing scripts that would reexport other datasets (such as Shin et al's dataset in our Table 1) to our CSV format, so that all our ML pipeline code could be applied to that data too.
> > >
> > > Note that some relevant recent fNIRS data releases (such as Mukli et al 2021, https://www.physionet.org/content/mental-fnirs/1.0/) do not use SNIRF format. SNIRF has admirable goals, but is not yet a de facto standard even among the BCI community. We believe it would not be especially accessible to the general ML audience we are trying to target. We believe our current format's *integration* with an ML prediction pipeline make it a reasonable solution for our intended users.
> > >
> > > Given your concern, we will reach out to the maintainers of SNIRF to inquire about possible tools that would accelerate making our dataset also available in that format. This may take time, we hope that is understandable. We don't know about how easily this format can integrate with the Python data science stack, but we will try to find out.

---

### Comment · Program_Chairs · 2021-10-13
**Official Ethics Review**

Because this is biometric information that could be perceived as sensitive, we recommend authors restrict distribution and monitor those making use of the dataset (ie. restricting access by request form, restricting possible use cases for the data, terms of use contract, etc.). Proposals such as a data license [1] may be a useful framework for thinking about how to approach this task.

There seems to be informed written consent, anonymization, participant compensation and reporting to the involved institutional review board (IRB). Authors also seem to have already reported demographic disparities (see Figure 1), and included an ethics review section (A.3), so we feel as though other issues have been acknowledged and adequately addressed.

[1] Benjamin, Misha, et al. "Towards standardization of data licenses: The montreal data license." arXiv preprint arXiv:1903.12262 (2019).

---

> ### Author Response · Authors · 2021-10-15
> **Clarifying questions about the request to restrict distribution and monitor use**
>
> Thank you for your careful review of our work. We are pleased that our main paper and supplement (especially discussion of potential negative impacts in Sec. A.3) have adequately addressed your standard concerns related to responsible care of human subjects data. We take our responsibility as stewards of this data seriously.
>
> Regarding the suggestion that our fNIRS dataset contains “biometric information that could be perceived as sensitive”, and thus we should “restrict distribution and monitor those making use of the dataset”, we would like to ask some clarifying questions:
>
> * Is the perceived risk that released fNIRS time-series sensor data (measurements of blood oxygen levels over time), perhaps together with other metadata, could somehow be used as a “fingerprint” to uniquely identify a human individual in our dataset?
> * Is there another perceived harm besides the above that is contributing to this suggestion?
> * Would this request for stricter terms be applied fairly to other similar datasets at this venue?
>
> In our view, the risk that our fNIRS data could be used to reidentify subjects and somehow cause harm is minimal. These sensors are quite expensive, not widely available yet, and the value of reidentification is minimal (what would a bad actor have to gain by knowing which individual participated in a paid 1-hour experiment at our university?). Our institutional IRB seems to agree, as it has approved a broad release under our current open-access CC-BY-4.0 license. All subjects indicated written informed consent of our current public open-access release. None requested more limiting terms of use.
>
> There are other datasets similar to ours, including at least two papers already *accepted* at this NeurIPS 2021 Datasets track, which also involve deidentified sensor data about specific human individuals, but which do not seem to have been requested to change the terms of use of their released data to restrict distribution further than our proposed CC-BY-4.0 license. Examples are:
> * EEGEyeNet https://openreview.net/forum?id=Nc2uduhU9qa
>   * Contains brain recordings via EEG of several human subjects
>   * Released under a CC-BY-4.0 license, just like ours
>   * Accepted to the NeurIPS 2021 dataset track in round 1
> * Stanford Knee MRI dataset (SKM-TEA, https://openreview.net/forum?id=YDMFgD_qJuA)
>   * Accepted to the NeurIPS 2021 dataset track in round 2
>   * Contains several hundred knee MRIs, which might also be considered “biometric” and “potentially sensitive”
>   * Released under a custom license (https://aimi.stanford.edu/lera-lower-extremity-radiographs-2)
>   * All SKM-TEA data is accessible to anyone without requiring applying for access, at this Drive link: https://drive.google.com/drive/folders/14yg5Oqan2fp0AChXjNOoytHenJWlcOlO
> * The Mental fNIRS 1.0 dataset (https://physionet.org/content/mental-fnirs/1.0/)
>   * This is a recent (2021) dataset release of fNIRS dataset similar to ours
>   * Released under a CC-BY-NonCommercial-ShareAlike license, which is like ours in being broadly permissive of many use cases without restricting access except for excluding commercial use
>   * Made available on PhysioNet, a curated platform with deep experience in releasing biomedical datasets under more protective terms when needed
>   * Cited in the latest revision of our paper as Mukli et al within our Table 1
>
> If we did implement more restrictive terms of use than our current CC-BY-4.0 license (e.g. requiring users to apply for access, putting data behind a password-protected gate), we are concerned this would definitely make the dataset more expensive and difficult to maintain (staff need to approve each request and be sure of compliance) and would limit the scientific impact and broader impact of our work (as the terms of use or logistical barriers involved might dissuade some legitimate users from benefiting from the data).
>
> If the Ethics Review board would like us to change the terms of use to be more protective, we will do so. We simply ask that the reasoning for this policy be clearly articulated and applied uniformly to all similar datasets who meet “sensitive biometric” criteria at this venue.

---

### Decision · Program_Chairs · 2021-10-09

**Decision:**

Accept

**Comment:**

This submission puts forward the largest dataset of fNIRS recordings labelled by mental workload indicators. The paper is well written and the dataset is well documented. Furthermore, the authors include a benchmark and a comprehensive and commendable discussion of the auditability of the dataset and the importance of including a range of diverse participants in the data collection process. This work can have significant impact within the BCI community. For this reason I recommend acceptance.

Flagged for an additional ethics review because this dataset is collected from humans.